# Tumor-targeted top1 inhibitor delivery with optimized parp inhibition in advanced solid tumors: a phase i trial of gapped scheduling

Anish Thomas [1] ✉, Nobuyuki Takahashi [1], Lenka Oplustil O'Connor [2], Christophe E. Redon[1], Chirayu Mohindroo [1], Linda Sciuto[1], Lorinc Pongor[1], Keith T. Schmidt[3], Seth M. Steinberg[1], Mirit I. Aladjem [1], William Douglas Figg [4], Mark J. O'Connor [2] & Yves Pommier[1]

Despite mechanistic rationale for combining PARP inhibitors with topoisomerase I inhibitors, clinical use has been hindered by dose-limiting toxicities. We hypothesized that integrating tumor-targeted topoisomerase I inhibitor delivery with optimized PARP inhibitor scheduling could enable effective combination therapy while reducing toxicity. In this trial (NCT02769962), we combined CRLX101, a nanoparticle topoisomerase I inhibitor, with olaparib using a gapped dosing schedule. The primary objective was to determine the maximum tolerated dose. Secondary objectives were to evaluate pharmacokinetics, pharmacodynamics, overall and progression-free survival. Twenty-four patients with advanced solid tumors were enrolled. The maximum tolerated dose for CRLX101 was 12 mg/m² every two weeks and olaparib 250 mg twice daily on days 3-13 and 17-26. Pharmacokinetics were consistent with monotherapy of each agent, and γH2AX kinetics revealed elevated DNA damage with the combination treatment compared to CRLX101 alone, supporting mechanistic efficacy. Among 19 evaluable patients, 2 patients had partial responses, and 6 had stable disease. Median overall survival was 6.06 months, progression-free survival 2.34 months, and duration of response 7.95 months. The combination showed acceptable safety across dose levels. Targeted delivery of a topoisomerase I inhibitor and gapped scheduling allowed higher olaparib dosing, showing promising activity and supporting the strategy's potential to widen the therapeutic window of DNA-damage response inhibitors while reducing toxicity.

DNA damage response (DDR) refers to coordinated cellular mechanisms that prevent DNA damage accumulation, thereby maintaining genomic integrity[1]. DDR plays a central role in cancer initiation, development, and progression. It has long been known that many cancers have defects in the DDR machinery. Historically, these defects have been exploited in cancer treatment through conventional chemotherapy and radiotherapy, although such approaches often result in collateral damage to normal tissues and side effects. In recent years,

[1]Developmental Therapeutics Branch, Center for Cancer Research, National Cancer Institute, National Institutes of Health, Bethesda, USA. [2]AstraZeneca, Cambridge, UK. [3]Clinical Pharmacology Program, Center for Cancer Research, National Cancer Institute, National Institutes of Health, Bethesda, MD, USA. [4]Genitourinary Malignancies Branch, Center for Cancer Research, National Cancer Institute, National Institutes of Health, Bethesda, USA. ✉e-mail: anish.thomas@nih.gov

the emergence of small molecule inhibitors targeting specific DDR components has enabled the preferential killing of cancer cells while minimizing impacts on normal cells, particularly in a synthetic lethal context[2]. As a result, DDR inhibitors are now being extensively explored in clinical trials to exploit these defects and enhance the efficacy of chemotherapies.

Poly (ADP-ribose) polymerase (PARP) inhibitors were the first DDR-targeted agents to receive approval for cancer therapy. PARP inhibitors selectively target PARP proteins, which are critical for the detection and repair of DNA damage, especially single-strand breaks (SSBs) through the base excision repair pathway[3]. The cytotoxic effects of PARP inhibitors largely stem from their ability to trap PARP-DNA complexes on endogenous DNA breaks, leading to the stalling or collapse of replication forks and the generation of deleterious double-strand breaks (DSBs)[4]. In replicating cells, DSBs are typically repaired via the homologous recombination pathway, which allows replication to continue. Cells harboring homologous recombination deficiencies – such as those with BRCA1 or BRCA2 mutations – are particularly susceptible to PARP inhibitors, leading to cell death from unrepaired DSBs[5,6]. Currently, PARP inhibitors are FDA-approved for breast, ovarian, prostate, and pancreatic cancers, and are being investigated for other malignancies[7].

Beyond their genotype-specific selectivity based on synthetic lethality, PARP inhibitors also enhance the efficacy of DNA damage-inducing chemotherapies[3]. Early studies showed that PARP1 deletion sensitizes cells to alkylating agents and topoisomerase I (TOP1) inhibitors, prompting the development of PARP inhibitors to potentiate the cytotoxic effects of therapy-induced single-strand breaks[8,9–12]. A central mechanistic rationale for combining TOP1 inhibitors with PARP inhibitors lies in their convergence on TOP1-DNA cleavage complexes (TOP1ccs). TOP1 inhibitors, such as camptothecin and its derivatives, trap TOP1ccs on DNA during replication, generating single-strand breaks that can convert into cytotoxic double-strand breaks if not properly repaired[13]. PARP plays a key role in the repair of these lesions through its involvement in the base excision repair pathway. The inhibition of PARP not only prevents repair of TOP1-induced DNA damage but also exacerbates replication-associated stress by trapping PARP1 on DNA, further destabilizing replication forks. Notably, this mechanism is independent of homologous recombination status, making the combination potentially effective across a broader patient population[9–12].

However, despite promising preclinical data, the clinical application of chemotherapy-PARP inhibitor combinations has encountered challenges, particularly dose-limiting toxicities (DLTs) such as myelosuppression[14–21]. For instance, a study by Kummar et al. found significant myelosuppression when combining the PARP inhibitor veliparib with topotecan, necessitating dose reductions for both agents[17]. The maximum tolerated doses (MTDs) of veliparib and topotecan represented only 3% and 40% of their respective single-agent MTDs. Clinical studies indicate that lower doses of PARP inhibitors may lead to reduced antitumor activity[22,23].

With the growing availability of potent and specific DDR-pathway small-molecule inhibitors (such as ATM, ATR, WEE1, DNA-PK, and others), pharmacological inhibition of DDR in patients is an area of intense study. The clinical use of DDR-pathway drugs, specifically addition of such agents to standard chemotherapy regimens has remained a challenge. The above considerations based on the PARP inhibitor clinical experience, point to a real need to better identify strategies that can lead to enhanced anti-tumor efficacy with DDR inhibitor-chemotherapy combinations while mitigating the unacceptable normal tissue toxicities.

We hypothesized that a dose-escalation strategy that incorporates tumor targeted DNA-damaging chemotherapy delivery alongside dose scheduling of PARP inhibitors could facilitate administration of effective PARP inhibitor-chemotherapy combinations[24,25]. To test this, we conducted a clinical trial combining olaparib with CRLX101, a nanoparticle formulation of camptothecin (CPT)[26,27], in patients with advanced solid tumors. CRLX101 exhibits prolonged circulation time and preferential tumor accumulation, offering improved safety compared to conventional TOP1 inhibitors such as topotecan and irinotecan[28–30].

The timing and duration of olaparib dosing were guided by preclinical studies in rat models, which more accurately mimic human hematopoietic DNA repair dynamics than traditional murine systems[25,26,31]. These studies demonstrated that (i) 24 h or more between CRLX101 and olaparib mitigated bone marrow toxicity and (ii) extended olaparib dosing post-CRLX101 enhanced efficacy without increasing toxicity[25,26]. Based on these insights, the clinical trial implemented a 48-hour delay between CRLX101 and olaparib to introduce an added safety buffer and ensure adequate marrow recovery (Fig. 1).

Here, we report results from the dose escalation phase of this clinical trial, detailing safety, preliminary efficacy, and translational correlates. This strategy – combining tumor-targeted drug delivery with schedule-based PARP inhibition – enabled dose escalation of both olaparib and CRLX101 to levels previously unattainable in clinical settings, resulting in preliminary clinical activity. Our findings provide a framework for overcoming the toxicity barriers of DDR-chemotherapy combinations by leveraging drug delivery and scheduling strategies to achieve a favorable therapeutic index.

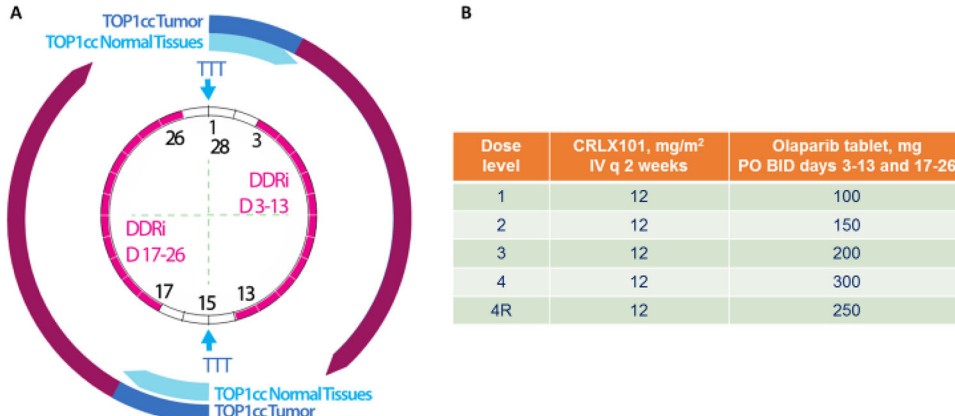

| Dose level | CRLX101, mg/m² IV q 2 weeks | Olaparib tablet, mg PO BID days 3-13 and 17-26 |
|---|---|---|
| 1 | 12 | 100 |
| 2 | 12 | 150 |
| 3 | 12 | 200 |
| 4 | 12 | 300 |
| 4R | 12 | 250 |

**Fig. 1 | Drug combination strategy and dose-escalation schedule. A** Drug combinations based on tumor-targeted chemotherapy delivery and protracted DDR inhibition, illustrated by the larger circle representing sustained chemotherapy exposure and the smaller circle indicating continuous PARP inhibitor activity. **B** Dose-escalation schedule TTT, Tumor Targeted TOP1 inhibitor; DDRi, DNA Damage Response inhibitor; TOP1cc, Topoisomerase I cleavage complexes.

**Table 1 | Patient characteristics**

| Characteristic | N = 24 (%) |
|---|---|
| Gender | |
| Female | 15 (63) |
| Male | 9 (37) |
| Median (range) age in years | 59 (48–76) |
| ECOG performance status | |
| 0 | 2 (8) |
| 1 | 22 (92) |
| Tumor type | |
| Non-small cell lung cancer | 4 |
| Small cell carcinoma | 3 |
| Pancreatic adenocarcinoma | 3 |
| Cholangiocarcinoma | 3 |
| Ovarian/ fallopian tube cancer | 3 |
| Cervical carcinoma | 2 |
| Colorectal carcinoma | 2 |
| Mesothelioma | 2 |
| Myxofibrosarcoma | 1 |
| Thymic carcinoma | 1 |
| Received prior systemic therapy | 24 (100) |
| Median number of prior therapies (range) | 3 (1–7) |

## Results

### Patient demographics

Between May 2016 and December 2017, 24 patients were enrolled (Table 1). All patients had received one or more prior lines of systemic therapy (Supplementary Table S1 and Supplementary Note 1) and had evidence of disease progression at enrollment. Eight patients (33%) had previously received a TOP1 inhibitor, while one patient (4%) had received a PARP inhibitor (Supplementary Table S1 and Supplementary Note 1). All patients received at least one dose of treatment and were evaluable for toxicities. Of these, 21 patients completed at least one cycle (median 2 cycles; range 1–7) and were evaluable for dose-limiting toxicities (DLTs). Three patients were not evaluable for DLT due to disease progression before completing cycle 1. In addition to these three, two other patients were not evaluable for response – one due to a cerebrovascular accident related to tumor embolism, and another due to prolonged neutropenia. One patient died while on treatment from an unrelated accident.

### Safety

There were no DLTs at DL1 ($n = 3$; olaparib 100 mg BID), DL2 ($n = 5$, 2 patients were not evaluable for DLT; olaparib 150 mg BID), or DL3 ($n = 3$; olaparib 200 mg BID). At dose level 4 (DL4, olaparib 300 mg BID), one patient experienced a DLT – grade 4 neutropenia that did not recover within 7 days. This cohort was expanded with 5 additional patients, with one additional DLT at this level due to a delayed treatment start (> 28 days) caused by an absolute neutrophil count < 1500. Consequently, DL4 was deemed intolerable.

Three patients were then enrolled at dose level 4 reduced (DL4R; olaparib 250 mg BID). One patient experienced a DLT – grade 4 neutropenia lasting more than 7 days. This cohort was expanded by 3 additional patients, none of whom experienced DLTs, establishing DL4R (CRLX101 12 mg/m² and olaparib 250 mg BID) as the maximum tolerated dose (MTD) and recommended phase 2 dose (RP2D).

The combination of CRLX101 and olaparib was generally well tolerated (Table 2). Across all dose levels ($N = 24$), the most common toxicities were anemia, leukopenia, and lymphopenia (71% each),

thrombocytopenia (58%), and neutropenia (54%). The most frequent grade 3 and 4 toxicities were leukopenia (46%), anemia (42%), neutropenia (38%), and thrombocytopenia (25%). One patient developed grade 3 febrile neutropenia. The most common non-hematologic toxicities were grade 1 or 2 fatigue (33%) and nausea (21%).

Dose interruptions due to treatment-related adverse events occurred in 11 patients, primarily for neutropenia recovery during cycle 1. Five patients required dose reductions, most commonly for neutropenia. Pegfilgrastim was not routinely used during the first cycle but was administered from cycle 2 onwards in five patients, cycle 3 in three patients, and cycle 4 in one patient.

### Pharmacokinetics

Pharmacokinetic (PK) parameters for CRLX101 and olaparib, based on plasma concentration-time profiles, are summarized in Supplementary Tables S2 and S3 and Supplementary Fig. S1 (Supplementary Note 1). After IV infusion of CRLX101, polymer-conjugated CPT concentrations increased rapidly, with a mean $C_{MAX}$ of 5230 ng/mL. Polymer-unconjugated CPT concentrations showed a gradual increase, with a mean $C_{MAX}$ of 266.9 ng/mL, suggesting controlled release of CPT from the polymer conjugate. Exposure to polymer-conjugated CPT (mean $AUC_{inf}$ of 168,737 hr·ng/mL) was nearly seven times higher than that of the unconjugated CPT (mean $AUC_{inf}$ of 25,418.6 hr·ng/mL). The nanoparticle had a mean terminal half-life of 47.42 hours, an estimated clearance of 0.117 L/hr, and a volume of distribution of 6.40 L, indicating retention in the circulation and well-perfused tissues. In a previous publication associated with this study, CPT distribution and release from the nanoparticle formulation were more closely examined using a population pharmacokinetic model[32]. Olaparib peak concentrations, observed 2 hours post-dose, increased proportionally with the dose. Importantly, PK parameters of CRLX101 and olaparib in this study were similar to those of individual agents administered by themselves[28,33,34].

### Efficacy

Among the 19 evaluable patients, two achieved confirmed RECIST-defined partial responses (PR), six had stable disease (SD), ten experienced progressive disease (PD), and one had clinical progression (Fig. 2A). Of the six patients treated at the MTD/RP2D, five were evaluable for response. Of these, two had ongoing PRs at the time of the data cut-off. One patient experienced a 15% reduction in target lesion size but died due to unrelated causes (accident), one had prolonged stable disease for over six months, and one had PD. The median duration of response was 7.95 months (95% CI 7.85–8.05). Notably, a patient with relapsed myxofibrosarcoma at DL4R had a deep and durable response lasting over seven months (Fig. 2B). This patient's tumor had a presumably deleterious *PALB2* K1124* mutation. Another patient with chemotherapy-refractory cholangiocarcinoma at DL4R also achieved a PR, maintained for six months. The median OS (Supplementary Fig. S2A) and PFS (Supplementary Fig. S2b) at a median follow-up of 30.2 months in the evaluable cohort ($n = 19$) was 6.06 months (95% CI 3.82–13.19) and 2.34 months (95% CI 1.91–5.53) (Supplementary Note 1).

### Pharmacodynamics

Hair follicles and PBMC were obtained pre-treatment on C1D1, pre-treatment on C1D3 (pre-olaparib), and C1D4 (24 h after olaparib) (Fig. 3A–G). γH2AX staining was used as a pharmacodynamic marker of DNA double-strand breaks. Following CRLX101 administration, an increase in γH2AX signal was observed on Day 3, indicating induction of DNA damage by CRLX101 alone. Notably, γH2AX levels were further elevated on Day 4 – after two doses of olaparib – relative to Day 3 in hair follicles and, to a lesser extent, in PBMCs. This pattern was observed across all dose levels (with limited evaluability in DL3), and supports the conclusion that the combination of CRLX101 and olaparib

**Table 2 | All treatment-Related Adverse Events (maximum grade, all cycles)**

| | DL1 (n = 3) | | DL2 (n = 5) | | DL3 (n = 3) | | DL4 (n = 6) | | DL4R (n = 7) | |
|---|---|---|---|---|---|---|---|---|---|---|
| **CRLX101 dose, mg/m2** | 12 | | 12 | | 12 | | 12 | | 12 | |
| **Olaparib dose, PO BID** | 100 | | 150 | | 200 | | 300 | | 250 | |
| | All gr | ≥ gr3 | All gr | ≥ gr3 | All gr | ≥ gr3 | All gr | ≥ gr3 | All gr | ≥ gr3 |
| Anemia | 2 | 0 | 2 | 1 | 2 | 2 | 5 | 1 | 6 | 4 |
| White blood cell decreased | 1 | 1 | 3 | 1 | 3 | 3 | 5 | 2 | 5 | 4 |
| Neutrophil count decreased | 1 | 0 | 1 | 1 | 3 | 2 | 4 | 3 | 4 | 3 |
| Lymphocyte count decreased | 1 | 0 | 3 | 0 | 2 | 1 | 6 | 1 | 5 | 3 |
| Platelet count decreased | 1 | 1 | 1 | 0 | 3 | 1 | 4 | 2 | 5 | 2 |
| Febrile neutropenia | – | – | – | – | – | – | – | – | 1 | 1 |
| Vomiting | – | – | 1 | 0 | 1 | 0 | – | – | 1 | 1 |
| Nausea | – | – | – | – | 2 | 0 | 1 | 0 | 2 | 0 |
| Dyspepsia | – | – | 1 | 0 | – | – | 1 | 0 | 1 | 0 |
| Anorexia | – | – | – | – | – | – | 2 | 0 | 1 | 0 |
| Bloating | – | – | 1 | 0 | – | – | – | – | – | – |
| Constipation | – | – | – | – | – | – | – | – | 2 | 0 |
| Diarrhea | – | – | – | – | – | – | – | – | 2 | 0 |
| Dysgeusia | – | – | – | – | – | – | – | – | 1 | 0 |
| Dysphagia | – | – | – | – | – | – | – | – | 1 | 0 |
| Alanine aminotransferase increased | – | – | – | – | – | – | – | – | 1 | 0 |
| Alkaline phosphatase increased | – | – | – | – | – | – | – | – | 1 | 0 |
| Aspartate aminotransferase increased | – | – | – | – | – | – | – | – | 1 | 0 |
| Hypophosphatemia | – | – | – | – | – | – | 2 | 0 | – | – |
| Hypokalemia | – | – | 1 | 0 | – | – | – | – | – | – |
| Hypomagnesemia | – | – | 1 | 0 | – | – | – | – | 1 | 0 |
| Alopecia | – | – | – | – | – | – | – | – | 3 | 0 |
| Fatigue | 1 | 0 | 1 | 0 | 1 | 0 | 2 | 0 | 3 | 0 |
| Sore throat | – | – | – | – | – | – | – | – | 1 | 0 |
| Pain in extremity | – | – | – | – | – | – | – | – | 1 | 0 |
| Sinus tachycardia | – | – | – | – | – | – | 1 | 0 | – | – |
| Allergic reaction | 1 | 0 | – | – | – | – | – | – | – | – |
| Infusion related reaction | 1 | 0 | – | – | – | – | – | – | – | – |
| Infusion site extravasation | 1 | 0 | – | – | – | – | – | – | – | – |

results in greater DNA damage than CRLX101 alone. Importantly, the increase from D3 to D4 occurred despite the same CRLX101 dose being used across cohorts, suggesting that the enhanced DNA damage was due to olaparib.

This additive effect was most pronounced at DL4R, the recommended Phase 2 dose, where olaparib was administered at its single-agent MTD (300 mg daily). A dose-dependent increase in γH2AX signal was observed across most dose levels, with the exception of DL3, where the signal decreased from D3 to D4, possibly due to limited evaluability or biological variability. γH2AX accumulation was more robust in hair follicles than in PBMCs, likely reflecting the replication dependence of these agents and the higher proliferative index of follicular keratinocytes.

**Genomic features associated with clinical outcomes**

In a post-hoc, exploratory analysis, we evaluated the genomic features associated with clinical outcomes to gain insights into possible drivers of response. Whole exome sequencing data of pre-treatment tumors were available from 11 patients, including 6 patients with matched tumor-normal pairs. None of the tumors demonstrating tumor regression had a *BRCA1* or *BRCA2* gene mutation likely to result in loss of function.

Given the role of DDR gene alterations, such as homologous recombination gene mutations in predicting responses to DDRi and

TOP1i, we analyzed patient tumors for somatic variants in a previously described set of 275 DDR genes[35] (Supplementary Fig. S3A and Supplementary Note 1). Among samples with tumor and normal sequencing available, we also generated a homologous recombination deficiency score[36]. We then assessed whether clinical benefit – defined as PR or SD lasting > 3 months – correlated with DDR gene mutations or homologous recombination deficiency status. Interestingly, the tumor with the highest homologous recombination deficiency score in the cohort demonstrated clinical benefit (Supplementary Fig. S3B and Supplementary Note 1). However, predicted loss-of-function variants (frameshift, stop, indel) in DDR genes were observed in both patients who experienced clinical benefit (50%, 2/4) and those who did not (71.4%, 5/7), showing no statistically significant correlation ($p = 0.73$). These exploratory findings suggest that HRD status alone may not fully explain the response to combined CRLX101 and olaparib treatment.

## Discussion

Our strategy of tumor-targeted delivery of chemotherapy, followed by the sequential administration of a DDR inhibitor (Fig. 1), allowed dose escalation of both the PARP inhibitor and TOP1 inhibitor to approximately 80% of their respective single-agent MTDs. This approach, modeled on preclinical studies examining the effects of chemotherapy and DDR inhibitors, achieved higher doses than what was feasible in previous clinical trials[14-21] (Fig. 4) (Supplementary Table S4 and

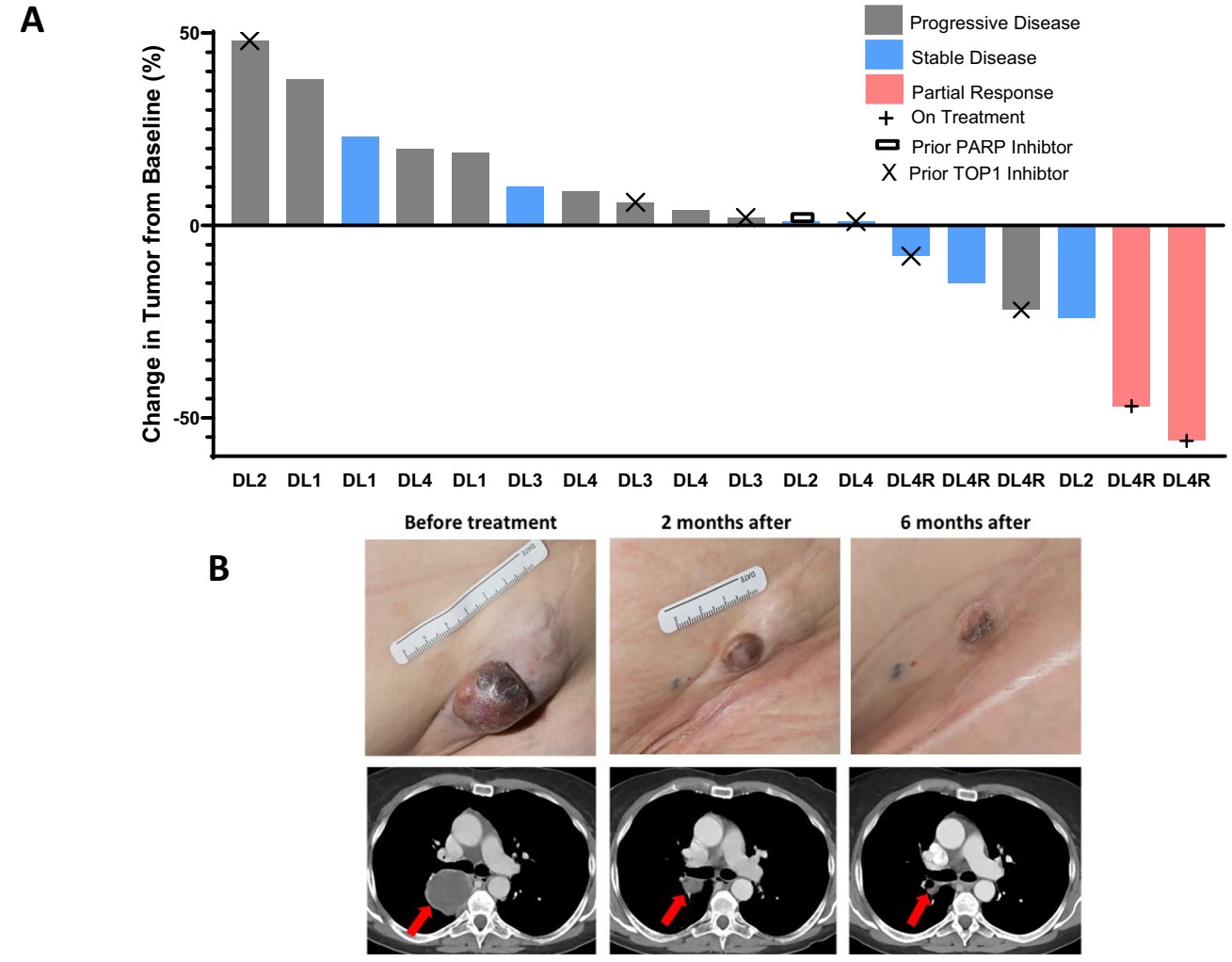

**Fig. 2 | Treatment Efficacy. A** Waterfall plot depicting tumor size changes in the evaluable cohort excluding the patient with clinical progression (*N* = 18 Patients). Source data are provided as a Source Data file. **B** Tumor response representative images from an individual patient demonstrating a partial response to the treatment.

Supplementary Note 1). We also observed encouraging clinical activity in heavily pretreated patients, providing proof of concept that this combination strategy could extend the therapeutic window of DDR inhibitors when combined with DNA-targeted chemotherapies. Consistent with this is the recently published phase II-III, randomized controlled

clinical trial of olaparib with carboplatin in neoadjuvant breast cancer patients that also used the same gap scheduling approach and achieved 100% overall survival[37].

This phase 1 clinical trial successfully identified the MTD for CRLX101 and olaparib, meeting its primary endpoint. The combination was shown to be feasible and tolerable, with adverse events largely consisting of myelosuppression, consistent with the known toxicities of both agents. Notably, the frequency of these events was lower than historical data from prior trials involving PARP inhibitor-chemotherapy combinations[14–21]. The most common toxicities were related to myelosuppression, which might have been further mitigated with prophylactic pegfilgrastim. Febrile neutropenia occurred in just one case, and non-hematological toxicities, such as fatigue and nausea, were generally mild to moderate.

PK studies confirmed that CRLX101 is associated with lower maximum concentration, longer elimination half-life, higher area under the curve, smaller volume of distribution, and slower plasma clearance than expected for camptothecin[38]. PK parameters

indicated no apparent interactions between CRLX101 and olaparib. Plucked hair bulbs, which contain actively replicating keratinocytes, served as a surrogate tissue to monitor in vivo DSB formation[34]. While acknowledging the limitations of surrogate tissues, we observed consistent γH2AX accumulation following CRLX101 administration, with further enhancement after olaparib exposure. Importantly, this increase occurred across dose levels despite a fixed CRLX101 dose, suggesting that olaparib augmented the DNA damage induced by CRLX101. While we cannot definitively exclude delayed effects of CRLX101 without a monotherapy arm, preclinical studies demonstrate that camptothecin-induced DNA damage is transient, typically peaking at 24–48 h and declining thereafter[25]. Thus, the sustained or increased γH2AX signal at 72 h (D4) is more consistent with an additive effect from olaparib. Although these pharmacodynamic assessments do not provide direct tumor-level confirmation, they offer supporting evidence for on-target activity and a cooperative mechanism of action, consistent with the synergy between TOP1 inhibition and PARP trapping.

Although anti-tumor activity was not the main endpoint of the study, we recorded encouraging signals at the MTD/RP2D without substantial additional toxicity. Of note, a patient with myxofibrosarcoma who had an ongoing deep and sustained response harbored a mutation in *PALB2*, which is involved in the recruitment of *BRCA2* and *RAD51* to DNA breaks in the homologous recombination pathway[39]. Although

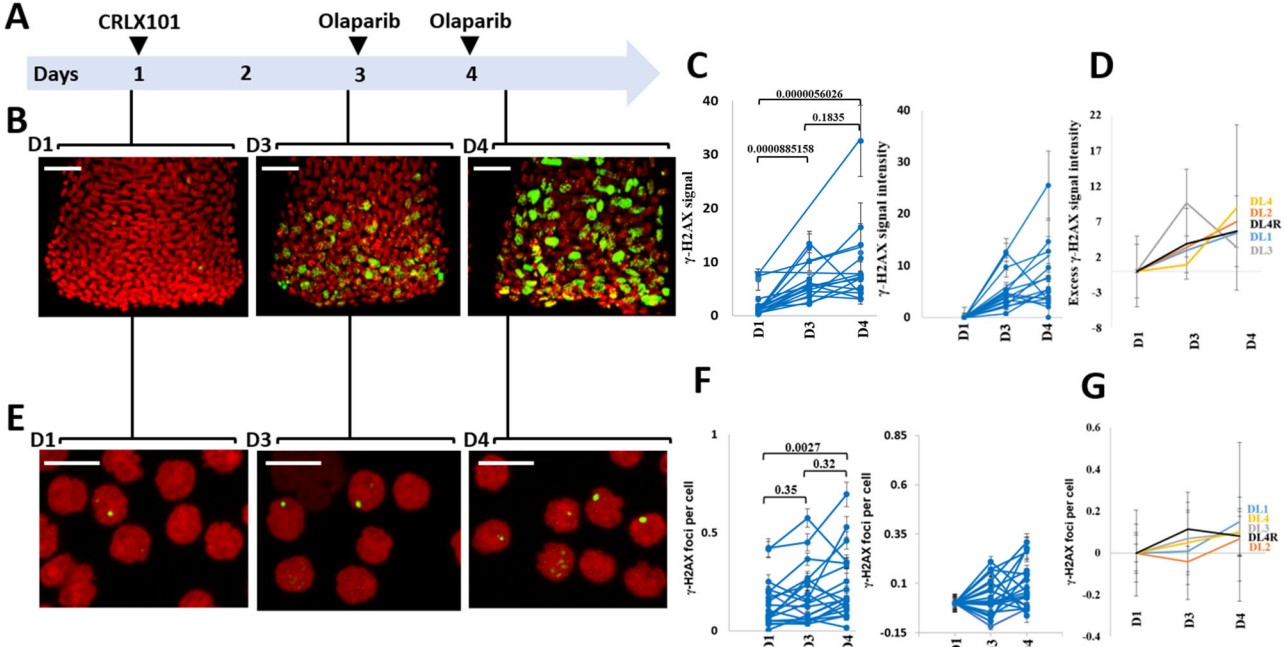

**Fig. 3 | Pharmacodynamic assessment of γ-H2AX formation in plucked hair bulbs and PBMCs following treatment with CRLX101 and olaparib. A** Diagram illustrating the timing of hair and blood collection: before treatment (day 1, D1), 48 h after CRLX101 and before olaparib (day 3, D3), and 24 h after olaparib (day 4, D4). **B** Representative images of γ-H2AX staining in plucked hairs at D1, D3, and D4 (original magnification, × 40). Scale bar = 50 µm. **C** Quantification of γ-H2AX in plucked hairs at D1, D3, and D4, shown as γ-H2AX intensities (left panel) and as excess γ-H2AX intensities after subtraction of background levels at D1 to represent treatment-induced γ-H2AX (right panel). Error bars represent the standard error of the mean (SEM) across hair per patient. The numbers of hairs analyzed were $N = 137$ hairs, $N = 115$ hairs and $N = 103$ hairs for D1, D2 and D3 respectively with 1, 20, 10, 11, 8, 2, 11, 2, 5, 2, 2, 8, 3, 3, 17, 10, 7, 5 and 10 hairs analyzed per patient respectively for D1; 7, 15, 19, 10, 8, 1, 5, 5, 4, 0, 0, 10, 6, 8, 9, 8, 9, 1 and 0 hairs per patient respectively for D2; and 3, 9, 7, 0, 10, 3, 6, 0, 0, 1, 4, 10, 8, 6, 12, 1, 11, 6 and 6 hairs per patient respectively for D3. D1 is the control (pre-infusion). Statistical analysis: A two-tailed Mann–Whitney test was used. Results were considered significant at $P < 0.05$. No adjustments were made for multiple comparisons. **D** Line graph depicting excess γ-H2AX intensities in plucked hairs with increasing doses of olaparib across different

dose levels. Error bars represent the standard deviation (SD) across patients. ($n = 19$ patient samples; $n = 1$ for DL1; $n = 5$ for DL2; $n = 3$ for DL3; $n = 5$ for DL4 and $n = 5$ for DL4R). Data are presented as mean values +/− SD. Source data are provided as a Source data file. **E** Representative images of γ-H2AX staining in PBMCs at D1, D3, and D4 (original magnification, × 63). Scale bar = 10 µm. **F** Quantification of γ-H2AX foci per cell in PBMCs at D1, D3, and D4, shown as γ-H2AX foci per cell (left panel) and as excess γ-H2AX foci per cell after subtraction of background levels at D1 (right panel). Blood samples were collected at D1, D3 and D4 from 20 patients ($n = 2$ patients for DL1; $n = 5$ for DL2; $n = 3$ for DL3; $n = 4$ for DL4 and $n = 6$ for DL4R). D1 represents the control group (pre-infusion). Error bars represent the SEM across PBMC per patient. Statistical analysis: For the D1 vs D4 comparison, a two-tailed Wilcoxon matched-pairs signed-rank test was used. For other group comparisons, a two-tailed Mann–Whitney test was performed. Results were considered significant at $P < 0.05$. No adjustments were made for multiple comparisons. **G** Line graph showing excess γ-H2AX foci per cell in PBMCs with increasing doses of olaparib across different DLs. Lines above groups indicate comparisons tested, with corresponding $p$-values shown. Error bars represent the SD across patients. Data are presented as mean values +/− SD. Source data are provided as a Source data file.

unrepaired endogenous DNA damage in HR deficient cells may sensitize them to PARP inhibition even in the absence of cytotoxic chemotherapy[2,4], such tumors may be even more sensitive to combinations of PARP inhibitors with DNA-damaging chemotherapy, and support the study of this combination in homologous recombination-deficient tumors.

Although our preclinical rationale posited HRD-independent synergy via convergence on TOP1 cleavage complexes, exploratory analyses revealed that one tumor with a high HRD score experienced clinical benefit. However, DDR gene mutations and HRD scores did not statistically correlate with response ($p = 0.73$), and responses were also observed in patients without known homologous recombination defects. These findings suggest that while HRD may contribute to sensitivity in select cases, it is not required for clinical benefit from this combination. Larger, biomarker-enriched trials are needed to refine predictive markers of response.

A key highlight of this study was the use of engineered drug carriers designed to extend circulation time, maximize tumor exposure, and minimize dose-limiting toxicities. Strategies to achieve this include tumor-targeted delivery of drugs via liposomes, PEGylation, and antibody-drug conjugates (ADCs)[24]. Liposomal and nanoparticle formulations, such as polymeric micelles, polymeric nanoparticles,

and liposomes, have shown promise. For example, nanoliposomal irinotecan (MM-398, Onivyde) has already been FDA-approved for pancreatic cancer, underscoring the clinical feasibility of this approach. However, clinical combinations with PARP inhibitors have highlighted formulation-specific considerations such as biliary elimination and gastrointestinal toxicity, which must be carefully optimized to achieve a favorable therapeutic index[21]. PEN-866, a miniature drug conjugate linking an HSP90 binder to an SN-38 payload, has also shown selective tumor accumulation and preliminary antitumor activity in a first-in-human study[40], further supporting the potential of rationally engineered carriers. PEGylated agents such as PLX038, a long-acting TOP1 inhibitor, are also in development. Clinical studies demonstrate early evidence of sustained antitumor activity in combination with PARP inhibitors, where the extended exposure to SN-38 enhances synergy with DNA repair inhibition. Similarly, SNB-101, a nanoparticle formulation of irinotecan, recently received FDA fast-track designation[41–43], reflecting growing interest in nanoparticle-based delivery systems.

Beyond nanoparticles, ADCs are rapidly emerging as a next-generation approach,[44,45] allowing higher doses of chemotherapy to be delivered specifically to tumor cells. Early clinical trials combining TOP1 ADCs with DDR-targeting agents such as ATR[46] and PARP

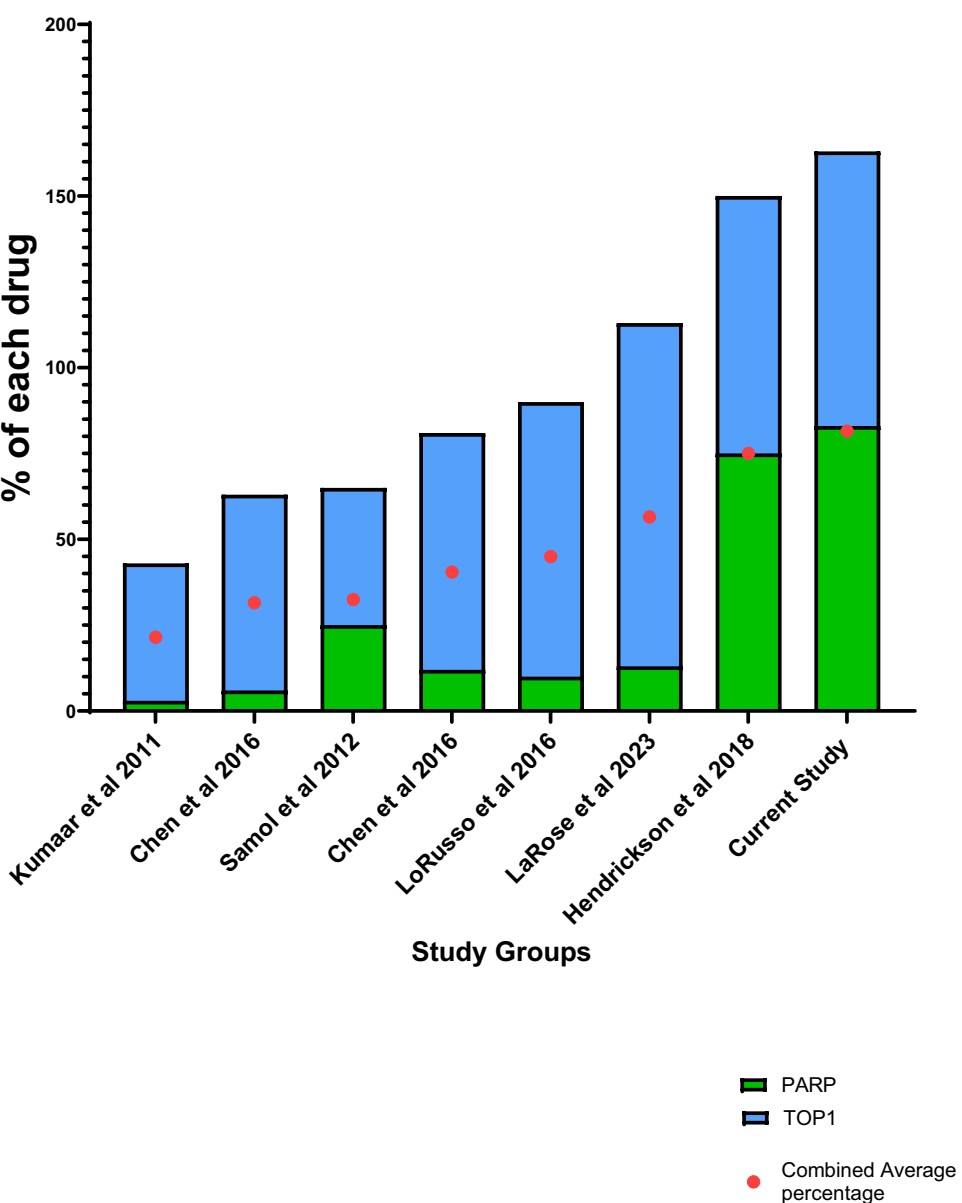

**Fig. 4 | Studies done using a TOP1 inhibitor and a PARP inhibitor.** Among all the studies (*N* = 8 studies) the combined dosage of was highest in the current study. Source data are provided as a Source Data file.

inhibitors[47,48] using a sequential dosing schedule have shown favorable safety and efficacy profiles, warranting further exploration. This strategy could be further refined with selective PARP1 inhibitors. Preclinical evidence suggests that PARP1 inhibition mediates the tumor-targeted effects of PARP inhibitors, while PARP2 inhibition is primarily responsible for myelosuppression[49]. AZD5305, a selective PARP1 inhibitor currently in early development, has shown promising efficacy with an excellent safety profile[50], and both this and another PARP1-selective agent, AZD9574[51] are being tested in combination with TOP1 ADCs in the clinic (NCT04644068; NCT05417594).

A key limitation of our study is the absence of on-treatment tumor biopsies, which would have provided more direct evidence of pharmacodynamic effects in the tumor microenvironment. Given the early-phase nature of the trial and the advanced disease stage of the enrolled population, paired biopsies were not mandated in order to minimize patient risk. Instead, we leveraged surrogate tissues (plucked hair follicles and PBMCs) to monitor DNA damage induction via γH2AX. While informative, these surrogate markers cannot fully recapitulate

tumor-intrinsic responses, underscoring the importance of incorporating tumor biopsies into future trials evaluating tumor-targeted DDR-based therapies. Additionally, while our results suggest a dose-dependent increase in γH2AX signal following the addition of olaparib, these findings should be interpreted with caution. The absence of a CRLX101 monotherapy control arm limits our ability to definitively attribute the observed γH2AX increase to olaparib. Furthermore, inter-patient variability, particularly at dose level DL3, adds complexity to the interpretation. As such, these pharmacodynamic findings should be considered exploratory.

Unanswered questions include the role of homologous recombination status in predicting response to this combination and the long-term toxicity effects. Since PARP inhibitors disrupt base excision repair, which helps repair chemotherapy-induced damage[2,3,8], their addition is expected to enhance the efficacy of these chemotherapies[12]. The ongoing phase II study will explore these effects in greater detail, focusing on the DDR mutations and homologous recombination status of tumors.

## Methods

### Study design and participants

We undertook a phase I/II study of CRLX101 and olaparib in patients with advanced solid tumors. Eligible patients were age 18 years or older and had histologically or cytologically documented unresectable, locally advanced, or metastatic solid tumors that were refractory to standard therapy and/or for whom no further standard therapy was available; Eastern Cooperative Oncology Group Performance Status of 0, 1, or 2; and adequate hematologic, renal, and liver function. CRLX101 and olaparib were supplied under a Collaborative Research and Development Agreement among the National Cancer Institute (NCI), Bluelink (previously provided by Cerulean), and AstraZeneca.

The study was conducted in accordance with the protocol, good clinical practice standards, and the Declaration of Helsinki and the International Conference on Harmonization. This study was conducted under an NCI Center for Cancer Research-sponsored investigational new drug application with institutional review board approval. All enrolled patients provided written informed consent before undergoing study-specific procedures. The trial was prospectively registered, and patient recruitment is complete. All primary and secondary objectives of the phase I study are reported in this work. The study protocol is available in the Supplementary file (Supplementary Note 2).

### Study treatment

This is an ongoing, open-label, phase I/II, dose escalation, and expansion trial of CRLX101 (nanoparticle CPT) plus olaparib conducted at the Center for Cancer Research, NCI. Here we present the results of the phase I dose escalation portion of the trial.

We hypothesized that the nanoparticle formulation of the TOP1 inhibitor (CRLX101) would selectively target tumor cells, reducing systemic toxicity and creating an optimal window for the sequential administration of a PARP inhibitor (olaparib), thereby enhancing the overall therapeutic efficacy[24]. To achieve this, olaparib was administered 48 hours after CRLX101 based on preclinical data, which showed that this window would allow for optimal sequencing and in vitro responses to DNA damage induced by IR and chemotherapy in human hematopoietic multipotent progenitors and mesenchymal stem cells, both components of bone marrow, and limit bone marrow toxicity[52]. Olaparib tablets were given at escalating doses across cohorts (as shown in Fig. 1), administered twice daily on days 3–13 and 17–26 of the treatment cycle. There was at least a 48 h window between olaparib and CRLX101. All patients across dose levels received a fixed dose of CRLX101 (12 mg/m² biweekly), delivered as a 1 h intravenous infusion on days 1 and 15 of a 28-day cycle.

CRLX101 was administered as a 1 h intravenous infusion on days 1 and 15 of a 28-day cycle. Normal saline (0.9% sodium chloride) was administered pre- and post-infusion of CRLX101 (1 liter over 2 h each). To reduce the risk of hypersensitivity reactions, patients received the following drugs 30–120 min prior to start of CRLX101 infusion: a corticosteroid (dexamethasone 20 mg IV), an antihistamine (diphenhydramine 50 mg PO) and an H2 antagonist (ranitidine 50 mg IV).

A standard 3 + 3 design was used for dose escalation. Starting doses of CRLX101 and olaparib tablets were 12 mg/m² and 100 mg, approximately 80% and 33% of their respective single-agent MTDs. DLTs were based on toxicities observed during the first cycle. DLTs were defined as: Grade 4 neutropenia complicated by fever ≥ 38.5 °C (i.e., febrile neutropenia) and/or documented infection; grade 4 neutropenia or thrombocytopenia that does not resolve within 7 days; any grade 3-4 thrombocytopenia complicated with hemorrhage; grade 4 anemia that does not resolve within 7 days despite optimal therapy (withholding study drug and red blood cell transfusions); inability to begin subsequent treatment course within 28 days of the scheduled date, due to study drug toxicity; any grade 3-4 non-hematologic toxicity (except fatigue/asthenia < 2 weeks in duration; mucositis in

subjects who have not received optimal therapy for mucositis; vomiting or diarrhea lasting less than 72 hours whether treated with an optimal anti-emetic or antidiarrheal regimen or not; or alkaline phosphatase changes). A new cycle of therapy did not begin until neutrophil count had recovered to >1500/mm³, platelets > 75,000/mm³, hemoglobin ≥ 8 mg/dL, and any toxicity recovered to ≤ grade 2, with no more than a 3-week delay permitted. Study treatment was discontinued if there was a > 3-week delay in reinstitution of treatment due to drug-related toxicity during a cycle. Dose reductions were allowed for toxicities. A maximum of two dose reductions were allowed. Dose re-escalation was not allowed.

### Study procedures

A history and physical examination were conducted at baseline and before each dose of CRLX101. Complete blood counts and serum chemistries were performed weekly during cycle 1 and every 2 weeks thereafter. Hematology evaluations were conducted more frequently upon observation of grade 2 or greater neutropenia or thrombocytopenia. Radiographic evaluation was performed at baseline and every two cycles for tumor response based on Response Evaluation Criteria in Solid Tumors (RECIST) version 1.1. Toxicities were graded using NCI Common Terminology Criteria for Adverse Events (version 4.0).

### Pharmacokinetic analysis

Blood samples for pharmacokinetic (PK) analysis were drawn at pre-dose CRLX101, mid-infusion (30 min post start), end of infusion (EOI), and 1, 2, 12, 24, and 48 h post EOI. Approximately 4 mL of blood was collected into a sodium heparin tube (BD Biosciences), immediately processed into plasma before storage in cryovials at − 80 °C. On C6D1, another set of blood samples for CRLX101 measurement was collected to assess if any drug interactions exist and to assess any CRLX101 accumulation. The 48 h post-EOI sample was also used to measure peak olaparib plasma concentrations (2 h post-oral administration). Total CPT, polymer-unconjugated CPT, and olaparib plasma concentrations were measured using previously published assays[53,54]. Polymer-conjugated CPT plasma concentrations were obtained via subtracting the polymer-unconjugated concentration from the total CPT concentration.

First dose pharmacokinetic parameters were calculated using noncompartmental methods (Phoenix WinNonlin 8.0, Certara Pharsight Corp, Cary, NC). Any plasma concentration measured below the LLOQ was excluded from analyses. The maximum plasma concentration ($C_{MAX}$) and time to $C_{MAX}$ ($T_{MAX}$) were recorded as observed values. The area under the plasma concentration vs time curve to the last observed time point ($AUC_{LAST}$) was calculated using the Linear Up Log Down trapezoidal rule. The elimination rate ($k_{EL}$) was calculated as the slope of the log-transformed concentrations vs terminal time points. AUC extrapolated to time infinity ($AUC_{INF}$) was calculated as $AUC_{LAST} + C_{LAST}/k_{EL}$, where $C_{LAST}$ is the concentration at the last observed time point. Half-life ($t_{1/2}$) was calculated as $\ln2/k_{EL}$. Apparent oral clearance (CL/F) was calculated as dose/$AUC_{INF}$; volume of distribution at steady-state (Vss) was calculated via the product of mean residence time extrapolated to infinity ($MRT_{INF}$) and last measured concentration ($C_{LAST}$). Pharmacokinetic parameters were calculated for each individual and summarized using the arithmetic mean ± standard deviation (SD).

### Pharmacodynamic analysis

PBMC and hair samples were obtained from all 24 patients on C1D1 (pre-treatment), on C1D3 (48 h after CRLX101 and pre-olaparib), and on C1D4 (24 h after olaparib). PBMCs were isolated by Ficoll gradient, washed 3 times with phosphate-buffered saline (PBS), fixed in 2% paraformaldehyde (PFA) for 20 min, washed 3 times with PBS, and spotted on slides by cytospin (800 rpm/4 min). PBMCs on slides were permeabilized with pre-chilled ethanol 70% and slides were stored at

4 °C overnight. PBMCs were then blocked for 30 min with 5% bovine serum albumin (BSA) in PBS-TT (PBS with 0.5% tween 20 + 0.1% triton X-100) before incubating 2 h with a mouse monoclonal anti-γ-H2AX antibody (Millipore Sigma cat# 05-636, clone JBW301) (dilution 500 in 1% BSA/PBS-TT) and then 1 h with a goat anti-mouse Alexa-488-conjugated IgG (ThermoFisher cat# A11029) (Dilution 500 in 1% BSA/PBS-TT). Finally, slides were incubated at 37 °C for 5 min with a solution containing RNAse A (0.5 mg/mL) and propidium iodide (PI) (5 μg/mL). Slides were then mounted with mounting medium containing PI (Vectashield, Vector Laboratories, Inc., Burlingame, CA, USA) and sealed with nail polish.

Hair bulbs were collected in 1.5 microtubes filled with cold PBS and placed on ice directly after plucking. Plucked hairs were then fixed for 20 min at room temperature with 2% PFA in 1.5 microtubes. Following 3 washes with 1 ml PBS, plucked hairs were imaged, and anagen hair bulbs hairs were selected for γ-H2AX detection. Samples were permeabilized with pre-chilled ethanol 70%, washed, and stored at 4 °C until use. γ-H2AX detection was performed as described for PBMCs, but times for blocking and staining procedures were increased by 50%. Washes were performed by 3 washes with PBS. After incubation with the RNAse and PI solution at 37 °C for 30 min, hair shafts were detached from the bulbs by using a razor blade, and hair bulbs were mounted between a slide and a coverslip with mounting medium containing PI and sealed with nail polish.

Samples were imaged by laser scanning confocal microscopy (Zeiss LSM 710 NLO). Optical sections through PBMCs and hair bulbs were combined in a maximum projection using the Zeiss Zen software. The foci were counted in PBMCs using the FociCounter software[55] by analyzing at least 200 cells, while γ-H2AX intensities were measured in the extremity of the hair bulbs with the Image J software. The above antibodies have been used in previous studies[56,57].

### Statistics & reproducibility

Descriptive statistics were used to summarize baseline patient characteristics, treatment-related adverse events, and PK and PD parameters. Comparisons of PK parameters between single-agent and combination treatments were performed using the Mann-Whitney U test for unpaired samples and the Wilcoxon signed-rank test for paired samples. Associations between genomic features and clinical outcomes were analyzed using the chi-square test. Survival curves were estimated using the Kaplan-Meier method. All analyses were conducted using GraphPad software version 10.1.1. Two-sided $p$-values less than 0.05 were considered statistically significant. No data were excluded from the analysis. The experiments were not randomized. The investigators were not blinded to allocation during experiments and outcome assessment.

### Reporting summary

Further information on research design is available in the Nature Portfolio Reporting Summary linked to this article.

## Data availability

Access to anonymized data is available upon reasonable request to qualified investigators for appropriate non-commercial use, with the expectation of contributing to a peer-reviewed publication. Requests should be directed to Dr. Anish Thomas, NCI/NIH, at anish.thomas@nih.gov. Approval will be subject to review by a regional ethics committee to ensure compliance with legal data processing requirements, data protection regulations, and ethical standards. Data access will be granted only for the purposes outlined in the data access agreement and for a duration sufficient to achieve those objectives. The study protocol is available in the Supplementary file (Note 2). Pharmacokinetic and pharmacodynamic data generated during the study are available to qualified investigators upon request, subject to

the same conditions and approvals outlined above. All other data supporting this work are included in the main article, Supplementary information, or source data file. Source data of Figs. 2A, 3C, 3D, 3F, 3G, Supplementary Figs. S1, S2A, S2B are provided with this paper. The data underlying Supplementary Fig. S3 are not publicly available due to patient confidentiality concerns related to the small, tumor-agnostic cohort. Access may be granted upon reasonable request, subject to data use agreements and institutional approvals as noted above. Source data are provided in this paper.

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

## Acknowledgements

This research was supported by the Intramural Research Program (NCI ZIA BC 011793) of the National Institutes of Health (NIH). The contributions of the NIH author(s) were made as part of their official duties as NIH federal employees, are in compliance with agency policy requirements, and are considered Works of the United States Government. However, the findings and conclusions presented in this paper are those of the author(s) and do not necessarily reflect the views of the NIH or the U.S. Department of Health and Human Services. AstraZeneca had no role in the data collection and analysis of this study.

## Author contributions

Conception and design: A.T., L.O.O., M.J.O. and Y.P. Collection and assembly of data: A.T., N.T., C.E.R., C.M., L.S., L.P., K.T.S. and S.M.S. Data analysis and interpretation: A.T., N.T., L.O.O., C.E.R., C.M., L.P., K.T.S., S.M.S., M.I.A., W.D.F. and M.J.O., Y.P. Manuscript writing: All authors. Final approval of manuscript: All authors accountability for all aspects of the work: All authors.

## Competing interests

The Authors declare the following competing interests: A.T. received grants to the NCI from EMD Serono Research and Development, Astra-Zeneca, Gilead Sciences, and ProLynx during the conduct of the study. L.O.O. and M.J.O. are full-time employees and shareholders at Astra-Zeneca. C.E.R. is a named inventor on a patent related to methods and kits (enzyme-linked immunosorbent assay) for measuring and quantifying DNA double-strand breaks using γH2AX and H2AX; however, this technique was not used in the present study. N.T., C.M., L.S., L.P., K.T.S., S.M.S., M.I.A., W.D.F. and Y.P. declare no competing interests.
