## [Transparent Peer Review file · Nature Communications]

Tumor-Targeted TOP1 Inhibitor Delivery with Optimized PARP Inhibition in Advanced Solid Tumors: A Phase I Trial of Gapped Scheduling

Corresponding Author: Dr Anish Thomas

Version 0:

Reviewer comments:

Reviewer #1

(Remarks to the Author)

This is a well-conducted phase I/II study of CRLX101, a nanoparticle TOP1 inhibitor, combined with Olaparib for treating patients with histologically or cyto-logically documented unresectable, locally advanced, or metastatic solid tumors that were refractory to standard therapy and/or for whom no further standard treatment was available. The study employed an appropriate design and methodology to estimate the maximum tolerable dose, assess the pharmacokinetics and pharmacodynamics of the doses, as well as the clinical endpoints.

I have the following suggested modifications:

1. Please report the individual concentrations over time as spaghetti plots as supplementary materials to justify the Cmax estimates provided in the main manuscript.
2. It would be informative to include the methods used for statistical analysis. Specifically,
 - a. What models were used for the pharmacokinetic and pharmacodynamic analysis
 - b. Figure S1 and elsewhere: Clarify if the error bars are based on standard errors or confidence intervals
 - c. How were the survival probabilities estimated? I assume based on Kaplan-Meier approach.
 - d. I assume in Figures 3C and 3F the numbers shown are p-values. Please specify the methods used to estimate these p-values. Note that there are extreme observations, and hence non-parametric approaches for assessing the difference will be appropriate.

Reviewer #2

(Remarks to the Author)

This article highlights a scientifically interesting and clinically pertinent question, namely maximizing/augmenting topoisomerase inhibitor-induced DNA damage by adding a PARP inhibitor. The authors seek to assess this, and maximize a therapeutic window and reduce systemic toxicity, by using CRLX101, a nanoparticle formulation of camptothecin with proposed preferential intratumor delivery, in combination with olaparib. Results from phase 1 dose escalation are presented, highlighting TRAEs, preliminary efficacy, and selection of the MTD/RP2D (DL4R). Additionally, correlative studies are presented for these dose escalation patients.

Major comments:

- Can the authors provide further data to justify the selection of the gapped dosing of olaparib (mandated to be given no less than 48 hours following CRLX101) on Days 3-13 and 17-26 of each 28-day cycle? This is only briefly alluded to under Results/Study Treatment (lines 232-236). A clearer understanding of the selection of the timing and duration of olaparib is needed.
- Acknowledging that this is a very heterogeneous patient population, it would be helpful to understand which patients may have had prior exposure to other topoisomerase inhibitors and PARP inhibitors. If there are patients who are top1i and/or PARPi-exposed, this would be of interest to notate on the waterfall plot.
- Regarding the pharmacodynamic data, were topo1 cleavage complexes (top1cc's) evaluated in addition to yH2AX? Top1CC's are referenced in the Figure 1A schema -- did this inform on the selection of the gapped olaparib dosing schedule, and was this assessed in PBMCs and hair follicles in enrolled patients? It would be helpful to readers to explicitly

address Top1CC's (in relation to the rationale for top1i+PARPi combo) within the introduction as well.

- Regarding the γ H2AX PD data presented, could the authors clarify if Figure 3C collates the γ H2AX changes in hair follicles across all dose levels? If so, does the γ H2AX data specifically for DL4R (selected MTD/RP2D) indicate a more dramatic change between D3 (CRLX101 effect alone) and D4 (CRLX101+olaparib effect)? Currently the conclusion noted in the body of the manuscript indicates that γ H2AX on D4 (CRLX101+olaparib effect) is greater than on D1 (pre-treatment), which is expected but not noteworthy if the intention is to demonstrate that doublet therapy (CRLX101+olaparib) augments DNA damage more than CRLX101 alone.

- An HRD score was assessed and is referenced as the one used in retrospective analysis across three trials in TNBC. It would be useful to note in which tumor types, of the patients enrolled, that this HRD score has also been assessed and/or is clinically validated (e.g. ovarian cancer).

Reviewer #3

(Remarks to the Author)

In this manuscript, the authors present data on a phase I clinical study of CRLX101, a nanoparticle-encapsulated preparation of the topoisomerase-1 inhibitor camptothecin with the PARP inhibitor olaparib in 24 patients with recurrent solid malignancies. The study population was fairly typical for this sort of study (median age 59, 10 different tumour types, median 3 prior lines of therapy). The dose of CRLX101 was fixed at 12 mg/m² on d1 and d15 of 28-day cycle based on phase I single agent data. Olaparib was given on days 3–13 and 17–26 (ie a 2-day gap before and after each dose of CLRX101) in escalating doses, starting at 100 mg bd. The design was a standard 3+3, somewhat disappointingly, with endpoints of safety/tolerability with PK and some limited normal tissue PD. There was some post-hoc genomic analysis.

Major comments.

I note that the study enrolled from May 2016 until December 2017, begging the question as to why it has taken over 7 years to be submitted for publication – this does suggest that the combination is perhaps not of the highest priority for onward clinical development.

The headline results show that it is not possible to give both agents at full dose (DL4) due to myelosuppression. However, a reduction in olaparib to 250mg bd (DL4R) was tolerable with only 1 DLT in 6 patients, allowing this dose to be specified as the RP2D. In terms of specific toxicity, as expected myelosuppression dominated with no other unexpected or G3+ toxicities.

In terms of PK, the C_{max} of unconjugated CPT was c.5% that of polymer-conjugated CPT, whilst steady-state volume of distribution is 6.65h, suggesting retention in the circulation – it is disappointing that no tumour biopsies were taken on study for this and PD analyses, especially given that the title of the manuscript includes the term 'tumor-targeted chemotherapy'.

The PD studies on hair follicles and PBMC demonstrate induction of DNA damage, as measured by γ H2AX fluorescence intensity, with a loose dose-dependent increase with increasing doses of olaparib and less damage in PBMC than hair follicles. This is where on-treatment tumour biopsies would have been very informative.

Finally, the post-hoc genomics do not really identify obvious genomic drivers. The text suggests that the patient presented in Figure 2B had a somatic PALB2 variant, although this is not presented in Figure S3A, where no variant in PALB2 is shown (unless I have missed it). There is, however, a RAD51D missense variant in one of the patients who benefitted. Conversely, BRCA2 frameshift variants were identified in two of the patients who did not benefit. 6 patients underwent HRD testing with the highest score identified in a patient deriving clinical benefit.

Overall, the study has been performed well, and the data are interesting. However, I am struggling with the novelty – as the authors state, PARP inhibitor combinations with top1 inhibitors have been extensively evaluated, with myelosuppression a major problem. I agree that this study shows that it can be possible to achieve near-full dose of both agents with careful scheduling and use of nanoparticles, but I am not sure that that is a major breakthrough in the field.

Clearly, in the 7 years since this study was completed, ADC have taken over as the method of choice for delivery of TOP1 inhibitors to tumours, and I am sure that TOP1-ADC/PARPi combination studies will be underway. The lack of on-tumour biopsies is disappointing – both for PK and (especially) PD – this is a significant shortcoming. The genomic assays are not terribly revealing but are important to exclude benefit limited purely to those with BRCA1/2 mutations.

Minor comment

What prior therapies had been given. Specifically, had any patient received prior Top1 or PARPi therapy? This is important information.

Version 1:

Reviewer comments:

Reviewer #1

(Remarks to the Author)

The authors have revised the manuscript to address my comments satisfactorily. Addition of statistical methods section will be beneficial to the readers to understand the results better.
I have no further comments.

Reviewer #3

(Remarks to the Author)

The authors have provided thorough responses to the reviewer comments and have modified the protocol. They also acknowledge areas of potential weakness in the study. Overall, I am happy with their rebuttal.

Reviewer #4

(Remarks to the Author)

Dr. Thomas and his colleagues reported the results of a phase I trial of the novel topoisomerase I inhibitor CRLX101 combined with olaparib. The trial was well-conducted, and the article is concise and well-written. The results demonstrated the feasibility of this combination treatment, possibly due to the nanoparticle formulation that prevents accumulation in the bone marrow. While this study provides some insight into the clinical development of PARP inhibitor and cytotoxic chemotherapy combinations, the provided pharmacodynamic data do not appear sufficient to prove the concept of combining Topo-1 and PARP inhibitors. Additionally, the finding that clinical benefit was associated with the HRD score contradicts the authors' hypothesis that Topo-1 and PARP inhibition work cooperatively, regardless of HRD status.

<Major Points>

1. Figure 3: The increase in γ H2AX from D3 to D4 may not be caused by the addition of olaparib on D3, but rather by the time course of the effect of CRLX101 itself. Is there any supporting data? Ideally, there would be γ H2AX time course data for CRLX101 alone.

2. I disagree with the description, "A trend toward increasing γ H2AX signal with escalating olaparib dose was observed," on lines 348–349 of page 9. Looking at Figure 3D, a decrease in γ H2AX is even observed in DL3 from D3 to D4. Additionally, I would like to see error bars for each plot to accurately understand the data.

Minor points:

1. The content of the Study Treatment paragraph in the RESULTS section is somewhat repetitive with the second paragraph in the METHODS section. I suggest that all of this information be included in the Methods rather than the Results.
2. Figure 2A: Does DL5 mean DL4R?
3. Is there a specific HRD score threshold used to distinguish HRD from non-HRD? It would be better to incorporate the HRD score for each patient in Figure S3A.

I reviewed the comments from the original reviewer (#2) and the authors' responses.

The authors addressed all of the concerns raised by Reviewer #2 except for one regarding the pharmacodynamic marker: The reviewer asked, "Regarding the pharmacodynamic data, were topo1 cleavage complexes (top1 cc's) evaluated in addition to γ H2AX?"

I agree with the reviewer's point that there are concerns about the robustness of the pharmacodynamic evaluation.

Version 2:

Reviewer comments:

Reviewer #4

(Remarks to the Author)

The authors tried to address the issues I raised in the last review. I appreciate their efforts. However, I still have concerns about the robustness of the pharmacodynamics studies. Although the authors argued that there is an olaparib-dose-dependent increase in γ H2AX intensity, the R2 value is still low at 0.42, even after removing DL3 patients. The lack of a CRLX101-alone control is critical and precludes the reliability of the pharmacodynamics data.

REVIEWER COMMENTS

Reviewer #1 (biostats):

This is a well-conducted phase I/II study of CRLX101, a nanoparticle TOP1 inhibitor, combined with Olaparib for treating patients with histologically or cytologically documented unresectable, locally advanced, or metastatic solid tumors that were refractory to standard therapy and/or for whom no further standard treatment was available. The study employed an appropriate design and methodology to estimate the maximum tolerable dose, assess the pharmacokinetics and pharmacodynamics of the doses, as well as the clinical endpoints.

We appreciate the positive feedback from the reviewer

I have the following suggested modifications:

1. Please report the individual concentrations over time as spaghetti plots as supplementary materials to justify the C_{max} estimates provided in the main manuscript.

We thank the reviewers for this suggestion and have now provided “spaghetti plots” to summarize individual conjugated and unconjugated CPT plasma concentrations over time in Figures S1B and S1C, respectively.

2. It would be informative to include the methods used for statistical analysis. Specifically,
a. What models were used for the pharmacokinetic and pharmacodynamic analysis

For pharmacokinetic analysis, the PK parameters reported were calculated using non-compartmental analysis (NCA). Via NCA, PK parameters were calculated for each individual patient and reported as mean values with standard deviation. This is now more clearly stated in the methods section. Table S2 has also been modified to report relevant total, conjugated, and unconjugated CPT PK parameter values, and updated to include additional cycle 6 data for 3 patients. A population pharmacokinetic (popPK) model using this dataset was previously published (Schmidt et al. Cancer Chemother Pharmacol. 2020 Oct;86(4):475-486.) and is now cited in the Results section.

b. Figure S1 and elsewhere: Clarify if the error bars are based on standard errors or confidence intervals

The error bars are in reference to standard deviation. This is now clearly stated for Figure S1A.

c. How were the survival probabilities estimated? I assume based on Kaplan-Meier approach.

These were estimated using the on Kaplan-Meier approach. We have now updated/clarified this in the Methods section under a new section Statistical Analysis.

d. I assume in Figures 3C and 3F the numbers shown are p-values. Please specify the methods used to estimate these p-values. Note that there are extreme observations, and hence non-parametric approaches for assessing the difference will be appropriate.

We have clarified in the figure legend that the reported numbers represent p-values. Additionally, we have updated the statistical analyses by applying non-parametric tests: the Wilcoxon signed-rank test for paired samples and the Mann-Whitney U test for unpaired samples. These changes have been reflected accordingly in the Methods section under Statistical Analysis.

Reviewer #2 (gynecological cancer chemotherapy, PARPi, clinical trial):

This article highlights a scientifically interesting and clinically pertinent question, namely maximizing/augmenting topoisomerase inhibitor-induced DNA damage by adding a PARP inhibitor. The authors seek to assess this, and maximize a therapeutic window and reduce systemic toxicity, by using CRLX101, a nanoparticle formulation of camptothecin with proposed preferential intratumor delivery, in combination with olaparib. Results from phase 1 dose escalation are presented, highlighting TRAEs, preliminary efficacy, and selection of the MTD/RP2D (DL4R). Additionally, correlative studies are presented for these dose escalation patients.

We thank the reviewer for their thoughtful comments and constructive suggestions. We have addressed each point below and revised the manuscript accordingly.

Major comments:

- Can the authors provide further data to justify the selection of the gapped dosing of olaparib (mandated to be given no less than 48 hours following CRLX101) on Days 3-13 and 17-26 of each 28-day cycle? This is only briefly alluded to under Results/Study Treatment (lines 232-236). A clearer understanding of the selection of the timing and duration of olaparib is needed.

The timing and duration of olaparib administration were informed by preclinical studies in rat models that more accurately reflect human hematopoietic DNA repair and toxicity profiles than traditional murine systems (O'Connor et al. bioRxiv, 2025, O'Connor et al. Mol Cancer Res (2017) 15 (4_Supplement): B32). These studies demonstrated that:

- A 24-hour delay between administration of CRLX101 and olaparib allowed sufficient bone marrow recovery from CRLX101-induced DNA damage, significantly reducing hematologic toxicity.**
- A prolonged duration of olaparib dosing (14 days vs. 2 days) following CRLX101 enhanced antitumor efficacy without exacerbating systemic toxicity.**

Importantly, rats were used instead of mice because murine hematopoietic stem cells preferentially utilize error-prone, non-homologous end joining pathways that are not representative of human DNA repair dynamics (Adams and Scadden. Nat Immunol. 2006 Apr;7(4):333-7.2010). In contrast, rat hematopoietic responses more closely parallel human marrow physiology, making them a more reliable model for predicting myelosuppression in early-phase trials.

Based on these findings, a 48-hour gap was selected for the clinical trial to introduce an added safety buffer beyond the 24-hour interval shown to be effective in preclinical studies. We took a conservative approach to account for interspecies variability in pharmacokinetics and marrow recovery kinetics, also aiming for a favorable therapeutic window. The extended olaparib dosing schedule (Days 3–13 and 17–26) was designed to maximize synergy while maintaining tolerability, based on efficacy data from the same preclinical work.

We have now revised the Results and Methods sections of the manuscript to better explain this rationale and have cited the relevant preclinical studies.

- Acknowledging that this is a very heterogeneous patient population, it would be helpful to understand which patients may have had prior exposure to other topoisomerase inhibitors and PARP inhibitors. If there are patients who are top1i and/or PARPi-exposed, this would be of interest to notate on the waterfall plot.

We appreciate the reviewer's comment regarding the prior systemic treatments received by the patients. In response to this, we have included an additional supplementary table (S1) detailing the prior treatment history of all 24 participants. Specifically, we have highlighted the number of patients who were treated with TOP1

inhibitors and PARP inhibitors on the waterfall plot. These details have also been incorporated into the manuscript in the 'Patient Demographics' section and Figure 2A.

- Regarding the pharmacodynamic data, were top1 cleavage complexes (top1cc's) evaluated in addition to γ H2AX? Top1CC's are referenced in the Figure 1A schema -- did this inform on the selection of the gapped olaparib dosing schedule, and was this assessed in PBMCs and hair follicles in enrolled patients?

While the formation of TOP1ccs represents a key mechanistic rationale for the combination strategy (as illustrated in Fig. 1A), we were not able to assess TOP1ccs in patients enrolled in the Phase I study due to limited tissue availability.

It would be helpful to readers to explicitly address Top1CC's (in relation to the rationale for top1i+PARPi combo) within the introduction as well.

We agree with the reviewer that explicitly addressing TOP1ccs would enhance clarity regarding the rationale for the TOP1 inhibitor and PARP inhibitor combination. Accordingly, we have revised paragraph 3 of the Introduction to include this mechanistic context, emphasizing the role of TOP1ccs in driving synergy between these agents.

- Regarding the γ H2AX PD data presented, could the authors clarify if Figure 3C collates the γ H2AX changes in hair follicles across all dose levels?

Yes the gamma-H2AX PD data presented in Figure 3C displays changes in gamma-H2AX in individuals at D1, D3 and D4 across all dose levels (DL1 100 mg Olaparib BID (n=1), DL2 150 mg Olaparib BID (n=5), DL3 200 mg Olaparib BID (n=3), DL4 300 mg Olaparib BID (n=5) and a level 4 reduced (DL4R 250 mg Olaparib BID (n=5)).

If so, does the γ H2AX data specifically for DL4R (selected MTD/RP2D) indicate a more dramatic change between D3 (CRLX101 effect alone) and D4 (CRLX101+olaparib effect)? Currently the conclusion noted in the body of the manuscript indicates that γ H2AX on D4 (CRLX101+olaparib effect) is greater than on D1 (pre-treatment), which is expected but not noteworthy if the intention is to demonstrate that doublet therapy (CRLX101+olaparib) augments DNA damage more than CRLX101 alone.

We agree that demonstrating the incremental effect of olaparib beyond CRLX101 alone is essential to support the rationale for the combination. In our analysis, we observed that γ H2AX levels were consistently higher on Day 4 (CRLX101 + olaparib)

compared to Day 3 (CRLX101 alone) across all dose levels, with the exception of DL3 (for which only one patient had Day 4 samples available). These findings support the interpretation that the observed increase in DNA damage is attributable to the additive effect of olaparib, as CRLX101 dosing was consistent across cohorts. We have revised the manuscript to explicitly state that the augmentation of γ H2AX signal on Day 4 relative to Day 3 likely reflects the contribution of olaparib to DNA damage beyond that induced by CRLX101 alone.

- An HRD score was assessed and is referenced as the one used in retrospective analysis across three trials in TNBC. It would be useful to note in which tumor types, of the patients enrolled, that this HRD score has also been assessed and/or is clinically validated (e.g. ovarian cancer).

FDA approval of HRD testing as a companion diagnostic for PARP inhibitor therapy is currently limited to ovarian cancer in our cohort, where it has been clinically validated and incorporated into treatment guidelines. In our cohort, only one patient had ovarian cancer, while the remaining five patients with available HRD scores had tumor types in which HRD is not currently validated as a predictive biomarker.

Reviewer #3 (cancer therapy, clinical trial, PARPi):

In this manuscript, the authors present data on a phase I clinical study of CRLX101, a nanoparticle-encapsulated preparation of the topoisomerase-1 inhibitor camptothecin with the PARP inhibitor olaparib in 24 patients with recurrent solid malignancies. The study population was fairly typical for this sort of study (median age 59, 10 different tumour types, median 3 prior lines of therapy). The dose of CRLX101 was fixed at 12 mg/m² on d1 and d15 of 28-day cycle based on phase I single agent data. Olaparib was given on days 3–13 and 17–26 (ie a 2-day gap before and after each dose of CLRX101) in escalating doses, starting at 100 mg bd. The design was a standard 3+3, somewhat disappointingly, with endpoints of safety/tolerability with PK and some limited normal tissue PD. There was some post-hoc genomic analysis.

Major comments.

I note that the study enrolled from May 2016 until December 2017, begging the question as to why it has taken over 7 years to be submitted for publication – this does suggest that the combination is perhaps not of the highest priority for onward clinical development.

The investigational agent CRLX101 underwent changes in ownership and development

priorities as it transitioned between companies, contributing to delays in publication. Although the commercial development of CRLX101 as a single agent was ultimately deprioritized, our findings support a strong biological rationale for combining tumor-targeted topoisomerase inhibition with PARP inhibition. These findings remain highly relevant in light of ongoing efforts to optimize DDR-chemotherapy combinations and address the persistent challenge of dose-limiting toxicities. With growing interest in next-generation delivery approaches, particularly ADCs, this work offers important proof-of-concept data that could inform future therapeutic strategies leveraging DNA repair vulnerabilities.

The headline results show that it is not possible to give both agents at full dose (DL4) due to myelosuppression. However, a reduction in olaparib to 250mg bd (DL4R) was tolerable with only 1 DLT in 6 patients, allowing this dose to be specified as the RP2D. In terms of specific toxicity, as expected myelosuppression dominated with no other unexpected or G3+ toxicities.

We thank the reviewer for summarizing this key finding. The full monotherapy dosing of both agents (DL4) was not feasible due to myelosuppression. However, we were encouraged to observe that a modest reduction in olaparib to 250 mg twice daily (DL4R) resulted in acceptable tolerability, with only one DLT in six patients. This allowed us to define DL4R as the RP2D. We also concur that the observed toxicity profile was consistent with known class effects: myelosuppression was the predominant adverse event, and no unexpected or grade ≥ 3 non-hematologic toxicities were reported.

In terms of PK, the C_{max} of unconjugated CPT was c.5% that of polymer-conjugated CPT, whilst steady-state volume of distribution is 6.65h, suggesting retention in the circulation –

We have now cited our previously published population-pharmacokinetic model developed using the present study, which further supports the retention of CPT in the circulation via the nanoparticle formulation (Schmidt et al. *Cancer Chemother Pharmacol.* 2020 Oct;86(4):475-486). The popPK model demonstrated distribution of both conjugated and unconjugated CPT into tissues but could not provide enough detail to distinguish uptake specifically into tumor tissue. While the present study did not directly assess uptake into tumor tissue, a prior study did demonstrate tumor uptake of CRLX101 (or more specifically CPT) in tumor tissue (Clark et al. *Proc Natl Acad Sci U S A.* 2016 Apr 5;113(14):3850-4.).

it is disappointing that no tumour biopsies were taken on study for this and PD analyses, especially given that the title of the manuscript includes the term ‘tumor-targeted chemotherapy’.

The PD studies on hair follicles and PBMC demonstrate induction of DNA damage, as measured by γ H2AX fluorescence intensity, with a loose dose-dependent increase with increasing doses of olaparib and less damage in PBMC than hair follicles. This is where on-treatment tumour biopsies would have been very informative.

We agree with the reviewer that on-treatment tumor biopsies would have provided valuable pharmacodynamic insight, particularly given the tumor-targeted nature of CRLX101. However, given the heavily pretreated, advanced-stage patient population and the phase I safety-focused design of the study, paired tumor biopsies were not mandated to minimize patient burden and procedural risk. Instead, we leveraged minimally invasive surrogates (plucked hair follicles and PBMCs) which demonstrated DNA damage induction (γ H2AX) with evidence of a dose-response relationship and tissue specificity. While these surrogate tissues do not replace tumor biopsies, they offer a practical and informative window into treatment-related pharmacodynamic effects in early-phase settings. We have revised the Discussion to acknowledge this limitation and the potential value of incorporating tumor biopsies in future trials evaluating tumor-targeted DDR-based therapies.

Finally, the post-hoc genomics do not really identify obvious genomic drivers. The text suggests that the patient presented in Figure 2B had a somatic PALB2 variant, although this is not presented in Figure S3A, where no variant in PALB2 is shown (unless I have missed it). There is, however, a RAD51D missense variant in one of the patients who benefitted. Conversely, BRCA2 frameshift variants were identified in two of the patients who did not benefit. 6 patients underwent HRD testing with the highest score identified in a patient deriving clinical benefit.

We thank the reviewer for their careful review of the genomic data. The patient presented in Figure 2B, who achieved a durable partial response, was found to have a somatic PALB2 mutation through clinical tumor sequencing performed outside of the trial, prompted by a strong family history of cancer. Unfortunately, germline tissue was not available for confirmatory testing, and the mutation is not shown in Figure S3A, which reflects only the genomic profiling data obtained within the study framework.

As the reviewer notes, additional DDR-related mutations, such as a RAD51D missense variant, were detected in another patient who derived clinical benefit. Conversely, BRCA2 alterations were observed in two patients who did not respond, underscoring the complexity of predicting therapeutic response based solely on individual gene

mutations. Similarly, while HRD testing was performed in six patients, only one had a high HRD score – and this patient experienced clinical benefit. These findings suggest that while DDR pathway alterations may enrich for sensitivity, benefit can still be observed in the absence of canonical HR-related mutations or high HRD scores.

As discussed in the revised Introduction, the rationale for combining TOP1 inhibitors and PARP inhibitors is mechanistically broad and not restricted to HR-deficient tumors. PARP inhibitors potentiate TOP1-induced DNA damage by blocking the repair of TOP1cc-mediated single-strand breaks and exacerbating replication-associated DNA stress. Therefore, this combination may have utility beyond BRCA-mutant or HR-deficient cancers, and our data support continued exploration of genomic and functional biomarkers that better capture DDR dependency.

Overall, the study has been performed well, and the data are interesting. However, I am struggling with the novelty – as the authors state, PARP inhibitor combinations with topo1 inhibitors have been extensively evaluated, with myelosuppression a major problem. I agree that this study shows that it can be possible to achieve near-full dose of both agents with careful scheduling and use of nanoparticles, but I am not sure that that is a major breakthrough in the field. Clearly, in the 7 years since this study was completed, ADC have taken over as the method of choice for delivery of TOP1 inhibitors to tumours, and I am sure that TOP1-ADC/PARPi combination studies will be underway. The lack of on-tumour biopsies is disappointing – both for PK and (especially) PD – this is a significant shortcoming. The genomic assays are not terribly revealing but are important to exclude benefit limited purely to those with BRCA1/2 mutations.

We agree that PARP inhibitor–TOP1 inhibitor combinations have a well-established rationale but have been historically limited by overlapping hematologic toxicity. Our study advances this field by demonstrating, for the first time to our knowledge, that tumor-targeted delivery of a TOP1 inhibitor via nanoparticle formulation (CRLX101) allows near-full dosing of both agents using a staggered administration schedule. While we recognize that delivery technologies have evolved, it provides important translational proof-of-concept with clinical relevance.

Indeed, the field has rapidly shifted toward ADCs as the leading strategy for targeted chemotherapy delivery with TOP1 inhibitors emerging as the fastest growing payloads of ADCs. However, published data on combining PARP inhibitors with tumor-targeted TOP1 inhibitors, whether via nanoparticle, liposomal, or ADC platforms, remain limited. We believe our study provides a framework (including dosing, safety, and pharmacodynamic benchmarks) that could directly inform the design of future ADC/PARPi trials.

We acknowledge the limitation of not obtaining on-treatment tumor biopsies for PK/PD analysis. To address this, we leveraged minimally invasive PD biomarkers (plucked hair follicles, PBMCs) to assess DNA damage dynamics in response to treatment. While not a substitute for tumor-based analyses, these surrogate tissues did demonstrate dose-related pharmacodynamic effects. Finally, while the genomic analyses did not identify clear predictors, they suggest that clinical benefit was not restricted to tumors with BRCA1/2 mutations. This aligns with the broader mechanistic rationale for combining PARPi with TOP1 inhibitors, which extends beyond homologous recombination deficiency and may apply to a wider subset of tumors.

In summary, although conducted prior to the ADC era, we believe this trial represents a meaningful clinical effort that remains relevant, particularly in its demonstration of tolerability and schedule-driven dose optimization. These findings can serve as a stepping stone for future strategies that seek to safely combine potent DDR inhibitors with targeted cytotoxic agents.

Minor comment

What prior therapies had been given. Specifically, had any patient received prior Top1 or PARPi therapy? This is important information.

We have now added this information in Supplementary Table S1, the Results section, and Figure S2A (waterfall plot).

Reviewer #1 (Remarks to the Author):

The authors have revised the manuscript to address my comments satisfactorily. Addition of statistical methods section will be beneficial to the readers to understand the results better.

I have no further comments.

Reviewer #3 (Remarks to the Author):

The authors have provided thorough responses to the reviewer comments and have modified the protocol. They also acknowledge areas of potential weakness in the study. Overall, I am happy with their rebuttal.

We thank the reviewers for their positive feedback.

Reviewer #4 (Remarks to the Author):

Dr. Thomas and his colleagues reported the results of a phase I trial of the novel topoisomerase I inhibitor CRLX101 combined with olaparib. The trial was well-conducted, and the article is concise and well-written. The results demonstrated the feasibility of this combination treatment, possibly due to the nanoparticle formulation that prevents accumulation in the bone marrow. While this study provides some insight into the clinical development of PARP inhibitor and cytotoxic chemotherapy combinations, the provided pharmacodynamic data do not appear sufficient to prove the concept of combining Topo-1 and PARP inhibitors. Additionally, the finding that clinical benefit was associated with the HRD score contradicts the authors' hypothesis that Topo-1 and PARP inhibition work cooperatively, regardless of HRD status.

We thank the reviewer #4 for their positive feedback. We clarify the following points regarding pharmacodynamic evidence and HRD status:

Pharmacodynamic evidence of synergy: While we acknowledge the limitations of using surrogate tissues (PBMCs and plucked hair follicles), the γ H2AX accumulation observed consistently across these samples, particularly in replicating follicular cells, demonstrated an additive DNA damage signal following olaparib administration on top of CRLX101-induced damage (please also see below response to Major point #1). Importantly, this effect occurred despite a fixed CRLX101 dose across cohorts, and γ H2AX levels correlated with increasing olaparib exposure. These findings provide mechanistic support, albeit indirect, that PARP inhibition potentiates CRLX101-induced DNA damage. We have clarified this point further in the revised Discussion.

Although our preclinical rationale posited HRD-independent synergy via convergence on TOP1 cleavage complexes, exploratory analyses revealed that one tumor with a high HRD score experienced clinical benefit. However, DDR gene mutations and HRD scores did not statistically correlate with response ($p = 0.73$), and responses were also observed in patients without known homologous recombination defects. These findings suggest that while HRD may contribute to sensitivity in select cases, it is not required for clinical benefit from this combination. Larger, biomarker-enriched trials are needed to refine predictive markers of response.

HRD status and clinical Benefit: We agree that our hypothesis postulated HRD-independent synergy based on PARP trapping at TOP1-induced lesions. Our exploratory analysis of HRD scores was intended to evaluate possible biomarkers of response, not to refute the mechanistic rationale. Indeed, clinical benefit was observed in both HRD-high and HRD-low tumors, including a patient with PALB2-mutated myxofibrosarcoma and a second with chemotherapy-refractory cholangiocarcinoma. While one patient with a high HRD score experienced benefit, others with predicted DDR mutations did not, and the correlation was not statistically significant ($p = 0.73$). We have now clarified in the Results and Discussion that this observation does not negate the potential HRD-independent mechanism of action, rather underscores the complexity of response determinants in vivo. We have clarified this point further in the revised Results.

Plucked hair bulbs, which contain actively replicating keratinocytes, served as a surrogate tissue to monitor in vivo DSB formation. While acknowledging the limitations of surrogate tissues, we observed consistent γ H2AX accumulation following CRLX101 administration, with further enhancement after olaparib exposure. Importantly, this increase occurred across dose levels despite a fixed CRLX101 dose, suggesting that olaparib augmented the DNA damage induced by CRLX101. Although these pharmacodynamic assessments do not provide direct tumor-level confirmation, they offer supporting evidence for on-target activity and a cooperative mechanism of action, consistent with the proposed synergy between TOP1 inhibition and PARP trapping.

<Major Points>

1. Figure 3: The increase in γ H2AX from D3 to D4 may not be caused by the addition of Olaparib on D3, but rather by the time course of the effect of CRLX101 itself. Is there any supporting data? Ideally, there would be γ H2AX time course data for CRLX101 alone.

We appreciate this important comment. As the reviewer points out, the observed increase in γ H2AX from D3 to D4 could, in theory, reflect delayed DNA damage from CRLX101 alone rather than an additive effect of olaparib. While we do not have a monotherapy arm of CRLX101 in this clinical trial to directly isolate its kinetic profile, we offer several points in support of olaparib's contribution:

1. **Fixed CRLX101 dose across cohorts:** All patients received the same dose and schedule of CRLX101, yet the D4 γ H2AX increase was more pronounced at higher olaparib doses (Figure 3D and 3G), indicating a dose-dependent effect from olaparib.
2. **Dose-dependent relationship:** As shown in Figure 3D and clarified in the updated analysis (see revised panel), γ H2AX signal intensity on D4 trended upward with increasing olaparib dose. This relationship became stronger when excluding DL3 (which included only a single patient at D4), yielding an improved correlation ($R^2 = 0.42$; right panel), suggesting the D4 increase is not purely a delayed CRLX101 effect but likely reflects a cumulative or synergistic response to olaparib.

3. **Time-resolved signal increase correlates with olaparib Administration:** The D3 to D4 interval (24 hours) aligns with the timing of two olaparib doses, suggesting a pharmacodynamic consequence of PARP inhibition rather than delayed kinetics of CRLX101, whose γ H2AX effects generally peak earlier based on preclinical modeling.
4. **Preclinical models:** In our rat and ex vivo human marrow models (O'Connor et al., bioRxiv 2025), we observed a plateau or decline in γ H2AX levels after the 48-hour mark post-CRLX101. These data support that further elevation of γ H2AX at 72 hours is unlikely to result from CRLX101 alone.

5. **Magnitude of change in high proliferative tissue: γ H2AX signal changes were especially robust in hair follicle keratinocytes (Figure 3B-D), a proliferative compartment that would be expected to exhibit olaparib-dependent PARP trapping effects during DNA replication.**

Nonetheless, we agree that without a CRLX101-alone control arm in patients, definitive attribution remains a limitation. We have now addressed this in the revised Discussion, clarifying the mechanistic inference and tempering our conclusions regarding proof of synergy.

While we cannot definitively exclude delayed effects of CRLX101 without a monotherapy arm, preclinical studies demonstrate that camptothecin-induced DNA damage is transient, typically peaking at 24-48 hours and declining thereafter. Thus, the sustained or increased γ H2AX signal at 72 hours (D4) is more consistent with an additive effect from olaparib.

2. I disagree with the description, "A trend toward increasing γ H2AX signal with escalating olaparib dose was observed," on lines 348–349 of page 9. Looking at Figure 3D, a decrease in γ H2AX is even observed in DL3 from D3 to D4. Additionally, I would like to see error bars for each plot to accurately understand the data.

We thank the reviewer for this observation. We agree that the description overstated the trend and have revised the text accordingly to avoid overinterpretation.

Specifically, we now describe the pattern as “a dose-dependent increase in γ H2AX signal was observed in most cohorts,” noting that DL3 did not follow this pattern, likely due to the small sample size for this cohort. While we collected hairs from 3 patients for D3, we got samples from one patient only for D4. Thus, the small sample size for D4 in this cohort may account for variability in pharmacodynamic response at that dose level. By comparison, we got 3 to 4 pairs D3/D4 for DL2, DL4 and DL4R.

We have updated Figure 3 to include error bars for each time point, showing standard deviation (SD) for group-level data per dose level and standard error of the mean (SEM) for individual patient-level data. These changes are reflected in the revised figure and its legend.

Minor points:

1. The content of the Study Treatment paragraph in the RESULTS section is somewhat repetitive with the second paragraph in the METHODS section. I suggest that all of this information be included in the Methods rather than the Results.

Thank you for this helpful suggestion. We have now combined the 2 paragraphs and included them in the methods section

2. Figure 2A: Does DL5 mean DL4R?

Yes, "DL5" in Figure 2A was a labeling error and should refer to DL4R, the reduced dose level of olaparib (250 mg BID) established as the recommended Phase 2 dose. We have corrected this in the revised figure and figure legend to maintain consistency throughout the manuscript.

3. Is there a specific HRD score threshold used to distinguish HRD from non-HRD? It would be better to incorporate the HRD score for each patient in Figure S3A.

We appreciate the reviewer's suggestion. As this was an exploratory analysis, we did not predefine a strict HRD score threshold to categorize patients as HRD-positive or HRD-negative. This decision was based on the limited sample size and the absence of a validated HRD cutoff in the tumor types represented in our cohort, with the exception of ovarian cancer. We have presented the individual HRD scores annotated by clinical benefit status in Figure S3B. Including these values again in Figure S3A would be redundant, as the data are already clearly visualized in S3B.

I reviewed the comments from the original reviewer (#2) and the authors' responses.

The authors addressed all of the concerns raised by Reviewer #2 except for one regarding the pharmacodynamic marker:

The reviewer asked, "Regarding the pharmacodynamic data, were topo1 cleavage complexes (top1cc's) evaluated in addition to yH2AX?"

I agree with the reviewer's point that there are concerns about the robustness of the pharmacodynamic evaluation.

As noted earlier in the review, although the formation of TOP1ccs constitutes a fundamental mechanistic rationale for the combination strategy (illustrated in Fig. 1A), assessment of TOP1ccs in patients enrolled in the Phase I study was not feasible due to limited tissue availability. On-treatment tumor biopsies would have provided pharmacodynamic insights, especially given the tumor-targeted design of CRLX101. However, owing to the heavily pretreated, advanced-stage patient population and the primary safety focus of this Phase I trial, paired tumor biopsies were not mandated to minimize patient burden and procedural risks. Instead, we employed minimally invasive surrogate tissues, plucked hair follicles and peripheral blood mononuclear

cells, which demonstrated induction of DNA damage, showing a dose-dependent response and tissue specificity. While these surrogate tissues cannot fully substitute for tumor biopsies, they provide a practical and informative means to evaluate treatment-related pharmacodynamic effects in early-phase clinical settings. This limitation has been explicitly acknowledged in the discussion section, underscoring the potential importance of including tumor biopsies in future studies of tumor-targeted DDR-based therapies.

REVIEWERS' COMMENTS

Reviewer #4 (Remarks to the Author):

The authors tried to address the issues I raised in the last review. I appreciate their efforts. However, I still have concerns about the robustness of the pharmacodynamics studies. Although the authors argued that there is an olaparib-dose-dependent increase in γ H2AX intensity, the R² value is still low at 0.42, even after removing DL3 patients. The lack of a CRLX101-alone control is critical and precludes the reliability of the pharmacodynamics data.

Response:

We thank the reviewer for their acknowledgment of our previous efforts. We agree that, despite the observed dose-dependent trend, the moderate R² value (0.42) underscores the variability inherent in pharmacodynamic biomarker studies and reflects the small sample size, particularly in dose level DL3. Most importantly, we acknowledge that the absence of a CRLX101-alone control arm in this clinical study limits the ability to conclusively ascribe the γ H2AX increase to olaparib administration.

In response, we have now further revised the Discussion section (see page 11, paragraph 7) to explicitly state this limitation and to more conservatively interpret the γ H2AX findings. We emphasize that while the results suggest an additive effect from olaparib, definitive attribution cannot be made without a monotherapy control. We also clarify that these data should be considered hypothesis-generating and warrant further validation in future studies designed with appropriate pharmacodynamic controls. The following lines have been added:-

" Additionally, while our results suggest a dose-dependent increase in γ H2AX signal following the addition of olaparib, these findings should be interpreted with caution. The absence of a CRLX101 monotherapy control arm limits our ability to definitively attribute the observed γ H2AX increase to olaparib. Furthermore, inter-patient variability, particularly at DL3, adds complexity to the interpretation. As such, these pharmacodynamic findings should be considered exploratory."